# Public School Food Supply Chain during the COVID-19 Pandemic: A Case Study of the City of Vitória (Brazil)

**Taniellen Miranda Coelho** [1], **Julianna Zambon Moscon** [1], **Irineu de Brito Junior** [1,2,*], **Angélica Alebrant Mendes** [3] **and Hugo Tsugunobu Yoshida Yoshizaki** [1,4]

1    Graduate Program in Logistics Systems Engineering, São Paulo University, São Paulo 05508-010, Brazil; taniellenmiranda@usp.br (T.M.C.); julianna.moscon@usp.br (J.Z.M.); hugo@usp.br (H.T.Y.Y.)
2    Environmental Engineering Department, São Paulo State University, São José dos Campos 12247-004, Brazil
3    Center of Engineering, Modeling, and Applied Social Science, Federal University of ABC, São Bernardo do Campo 09606-045, Brazil; angelica.alebrant@ufabc.edu.br
4    Department of Production Engineering, São Paulo University, São Paulo 05508-010, Brazil
*    Correspondence: irineu.brito@unesp.br; Tel.: +55-12-99702-9119

**Abstract:** *Background:* Due to the COVID-19 pandemic, Brazilian public schools closed in 2020. This lockdown stopped the provision of school meals to public school students, most of whom belonged to low-income families facing food insecurity. To guarantee the students' food security during this period, food items previously provided through school meals were converted into food kits and delivered to the students' families. *Methods*: This case study analyzes the logistical impacts of this change in the school food supply chain concerning the legislation, procurement, assembly, and distribution of food kits in the city of Vitória, Brazil. We interviewed suppliers and workers of the Municipal Secretariat of Education and distributed a survey to professionals and beneficiaries. *Results*: One of the findings was that federal procurement regulations for the acquisition of food for public schools led to difficult choices for school officials during this period. These regulations determined that at least 30% of the budget must be used in local purchases from smallholder family farmers. However, almost all products generated by family farming in the region of Vitória are perishable and require distribution and consumption on the same day, which represents a challenge for the logistic process of assembling and distributing food kits. The solution was the selection of eggs as the primary protein item in the kits. *Conclusions:* The lessons learned through this study suggest potential actions that would make this supply chain more resilient in future emergencies.

**Keywords:** school feeding; food security; COVID-19 pandemic; distribution; supply chain management; legislation

## 1. Introduction

The first case of COVID-19 was confirmed in Brazil on 26 February 2020 [1]. All public schools were closed on 17 March 2020 in the Brazilian State of Espírito Santo to maintain social isolation and quarantine as cases increased. Approximately 45 thousand students lost their school meals, which had been provided in public schools in the city of Vitória (with 365 thousand people, it is the capital of Espírito Santo State) prior to the pandemic. For many of these students, it was their only meal of the day [2]. Since the schools were closed, more than 39 billion meals were no longer available around the world [3].

According to UNICEF [4], the COVID-19 crisis has impacted Brazilians' income. Approximately 55% of the interviewed citizens confirmed that they experienced a reduction in their incomes during this time period. The reductions were larger in families with children and teenagers, with an approximately 63% income reduction. This survey also highlighted an increase in industrialized food consumption. Low-income families are at risk of food insecurity and negative health consequences during the COVID-19 pandemic without appropriate support [5,6].

In 2010, Brazil was one of the only three countries in the world that included the right to food consumption in its Constitution [3]. The students of Brazilian public schools used to receive at least one meal per day as part of the National School Meals Program (Portuguese Acronym: Programa Nacional de Alimentação Escolar—PNAE) [7]. School feeding is a key component of Brazil's strategy for addressing food security and nutrition, which links education, agriculture, health, and social protection for lower-income people [8,9].

PNAE has the objective of contributing to biopsychosocial growth and development, learning, school performance, and the construction of healthy habits in students through action toward food and nutritional education and the offer of meals that cover at least 15% of the daily nutritional necessities of students during their periods in school [10,11].

The program provides safe and healthy food, as food is acquired from local suppliers [7]. Therefore, at least 30% of program funds must be used in the direct purchase of products from smallholder family farmers [12]. Additionally, to support small producers and encourage economic and sustainable development in communities, this measure also increases access to fresh and nutritious food [3,9]. Between 2011 and 2017, only one-third of the Brazilian state's capitals met the established target of at least 30%, which demonstrates the need to encourage smallholder farming production and to better control assigned resources [13].

To guarantee students' food security during the pandemic, the government authorized cities to distribute food kits that met nutritional requirements [14] for families instead of school meals using PNAE funds [7]. After this authorization, the mayor of Vitória created a program for the purchase, assembly, and distribution of food kits for all public school students.

Many Brazilian cities consider buying food directly from smallholder family farming for school meals as a challenge because there are a limited number of producers to supply a large demand. In addition, the inclusion of farmers in the PNAE requires that they be organized into associations or cooperatives to sell products with a formal receipt to meet fiscal requirements [15,16].

Given this scenario, our research question is as follows: What were the supply chain challenges faced by the public-school food distribution system during the COVID-19 pandemic in Brazil? Thus, the main objective of this case study applied to the city of Vitória (Brazil) is to determine the difficulties in the acquisition, assembly, and distribution of food kits during the public schools' lockdown and the necessities for modifying processes and legislation. The lessons learned allow the suggestion of potential actions that would make this supply chain more resilient in future emergencies.

We mapped the process, conducted semi-structured interviews with professionals and food suppliers, and provided questionnaires to teachers, school workers, and parents. This study provides information about the logistics process, bottlenecks, failures, waste, risks, and public regulations.

After this introduction, in Section 2, we present the methodological approach. The results and their discussion are presented in Sections 3 and 4, respectively. Finally, in Section 5, we present the conclusions of the study.

## 2. Materials and Methods

The research method was a case study [17,18] with exploratory [19] and descriptive objectives with a qualitative and quantitative approach [20–22]. It was developed in the city of Vitória in Brazil through data collection and interviews with employees of municipal public schools, beneficiaries, and suppliers.

The case study methodology was adapted from the studies of Eisenhardt [17] and Miguel [18], and the steps are illustrated in Figure 1 below.

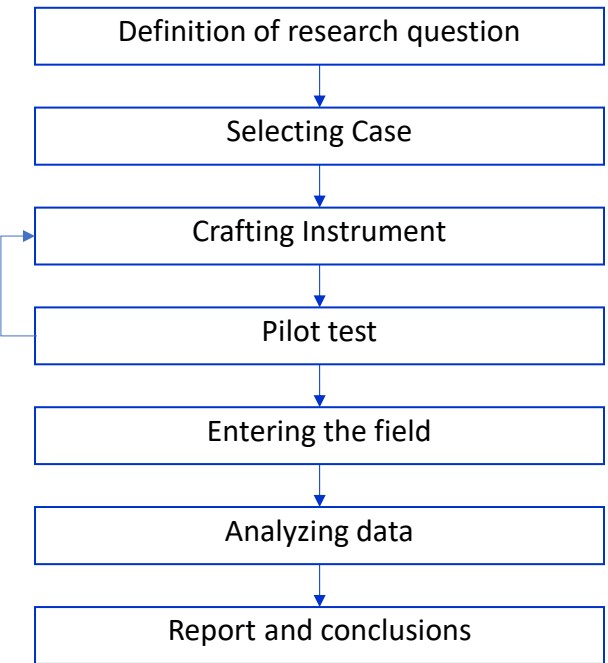

**Figure 1.** Case study methodology. Source: Adapted from [17,18].

- Definition of research question:

Prior to the pandemic, school meals were provided to students who needed them. The logistical process for providing these meals had long been established and was in steady operation. During the pandemic, these meals were replaced by food kits that were distributed to the students' families. This logistics change defined the abstract idea that the research aimed to measure using survey questions (construct) [23]. The research question was as follows: What were the supply chain challenges faced by the public-school food distribution system during the COVID-19 pandemic?

- Selecting the case:

The choice of the case of the Vitória City Education Network was due to the support of the authors by the Municipal Secretariat of Education (SEME) to identify the difficulties found in the supply chain for the distribution of food kits for students. Vitória is the capital of the Brazilian state of Espírito Santo, has a population of 369,534 inhabitants and a density of 3338.30 inhabitants/km², and is the main city of a metropolitan area consisting of 7 cities with more than 2 million inhabitants. In Vitória, 18% of the population is characterized as poor, and 12% is considered extremely poor [24]. In 2020, there were 45,331 students enrolled in 96 public schools in 8 regions spread throughout the city.

In Brazilian society, high-income and mean-income students usually attend private schools because it is known that private schools provide a better education. Public schools are usually attended by low-income students, for whom the school meals provided have relevant importance in terms of food security [25,26]. Figure 2 compares the monthly income of the city population and the public school students' families according to the Brazilian minimum wage (MW) (1 MW $\cong$ 227.4 USD). It can be observed that more than 65% of the public school students' families receive less than 2 MW monthly income. Additionally, most students of high-income families are not enrolled in public schools.

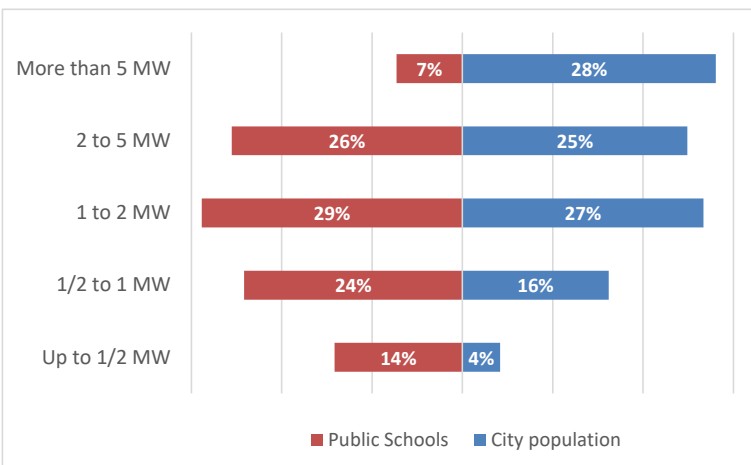

**Figure 2.** Comparison of family income of the public school students' families and the city population of Vitória. Source: Authors using adapted data from [27].

According to IBGE (Portuguese acronym—Brazilian Institute of Geography and Statistics) [27,28], 24.7% of Brazilian families are below the poverty line of USD 5.50 per day. This amount corresponds to less than four fifths of the Brazilian minimum wage per month (USD 165). Figure 2 shows that most of the students of these families are enrolled in public schools.

- Crafting instrument:

Initially, a document and bibliographic search was performed through document collection from government agencies and media, followed by a semi-structured interview with the SEME technical team, including managers, logistics coordinators, and nutritionists. After this, a questionnaire was sent to teachers, school workers, and students' parents or guardians.

During the pandemic, interviews were conducted online to understand the context of the acquisition, assembly, and distribution of food kits, to map the entire supply chain, and to understand the logistical flow of the parties (stakeholders).

Semi-structured interviews were conducted with the two main suppliers (of dry goods and smallholder family farming products) of SEME.

For the first company, Vila Vitória, the interview was conducted in November 2020, in addition to an on-site visit. The company is responsible for the purchase, assembly, and distribution of dry or nonperishable food to the school units. An interview was conducted with the person responsible for the logistics of the company to understand and elaborate the logistical flow of the process.

In the Cooperative CafSerrana, representing the deliveries referring to smallholder family farming and responsible for distributing protein (chicken eggs) to the schools, an online interview was conducted in December 2020, with the president of the cooperative, with the purpose of understanding and describing their logistical flow.

The elaboration of the questionnaires aimed to measure (i) program scope; (ii) the impact of the lack of meals on families; (iii) the satisfaction level of the beneficiaries; (iv) the duration and quality of the food kit; and (v) difficulties in collecting the kits.

Two different structured surveys were developed, one for professionals (teachers and employees) and the other for beneficiaries (students' parents or guardians). SEME sent the questionnaire links to all regions of the Vitória City Education Network.

- Pilot test:

Before sending the questionnaires to all school units in the network, a pilot test was conducted by providing the forms to parents and professionals from only one school, with constant monitoring by the school's management. After the pilot test, the necessary corrections and adjustments were made to improve the questionnaire [18]. Both questionnaires are shown in the Supplementary Materials.

- Entering the field:

The questionnaires were sent to all professionals and beneficiaries. The sampling method used was probabilistic; in other words, all individuals in the population had the same chance to participate in the sample.

The questionnaire was answered anonymously, and the first question was regarding acceptance of the informed consent form (ICF). The survey complied with all research-with-human rules from our institution.

Regarding the return of the questionnaires, a total of 45,331 students enrolled, 2674 responses (6%) were received, and the calculated margin of error was 1.84% with a 95% confidence level. For the 510 professionals involved in the distribution, 222 responses (44%) were received, with a margin of error of 4.95% and a confidence level of 95% [29].

Concerning the sample of the questionnaires sent to professionals, we received answers from professionals from all the school zones, as shown in Figure 3. It is worth mentioning that different zones may have different quantities of public schools and, consequently, different amounts of professionals.

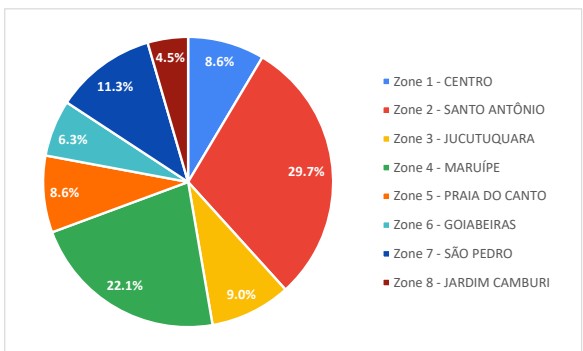

**Figure 3.** Amount of professional questionnaire answers from each school zone.

For the sample from students, questionnaires did not collect information about school zones because, according to the SEME managers, most of the parents or guardians may not be able to locate their school correctly. Additionally, it was not our objective to collect information about income, gender, or age since we know the income family profile of public schools (Figure 2). Additionally, we were told by SEME that families of higher incomes that had children enrolled in public schools did not require the kit. Consequently, these questions were not an issue to our research.

It is important to highlight that the answers for both questionnaires collected attribute data to make the answers simple for people who may not have had the opportunity to obtain a high level of education. In this sense, data distribution probabilities are not possible to make. Please see both questionnaires in Supplementary Materials.

- Analyzing data and reporting and conclusions:

In the last stage of the research, the information obtained from the interviews and questionnaires was described, compiled, and analyzed. This is detailed in the Results and Discussion sections.

## 3. Results

The results are divided into three parts: assembly of the kits, purchase and distribution of food kit items, and results of the questionnaires.

### 3.1. Assembling the Kits

Food previously provided in the form of meals was served at the schools. Many items were fresh and perishable, with a short shelf life. These items had to be replaced by food items with a longer shelf life. One of the greatest logistical difficulties was the supply of

protein. Before the pandemic, meats were served. During the pandemic, there was a need to replace the way in which the protein was supplied due to the problem of providing food items that required refrigeration.

The first step of the food kit distribution project was to define which items would compose the food kit and which quantities aligned with nutritional requirements [30]. Nutritional planning was performed by a team of nutritionists based on the menu offered by the schools, and families were informed of the need to supplement their food with other food items.

The estimated needs for macronutrients and energy expenditure depend on age and gender and vary mainly according to body size and physical activity [31]. The estimated nutrient requirements are 117 kcal/kg body weight from 0 to 0.5 years of age, 108 kcal/kg from 0.5 to 1.0 years, and 1300 kcal from 1 to 3 years. In boys between the ages of 10 and 20 years, the estimated energy intake is 2500 to 3200 kcal/day, and girls require approximately 1800 to 2000 kcal/day [32].

Thus, considering the students' ages and 50% boys and 50% girls, the weighted average caloric intake of a student aged 3 to 18 years in Vitória is 1885 kcal/day.

Because of the different age groups of beneficiaries (babies, children, youngsters, and students with special needs and dietary restrictions), the major challenge for nutritional planners was to measure the ideal amount of each food item for the kits. In addition, it would not be recommended (due to high risk of food contamination) to fractionate some items since the amounts calculated per beneficiary are incompatible with the amounts sold per individual commercial packaging [33]. For example, the standard vegetable oil container holds 0.9 L. It is not possible to distribute less than 0.9 L per kit.

In accordance with nutritional criteria, and the quantities consumed by students in conventionally sold packages, the kits were assembled with the items described in Table 1 for 90 days.

**Table 1.** Products included in the food kit.

| Item Provided | Amount per Kit | Carbohydrate (g) | Protein (g) | Total Fat (g) | Saturated Fat (g) | Fiber (g) | Sodium (mg) | Kcal |
|---|---|---|---|---|---|---|---|---|
| Rice | 2 kg | 1576 | 144 | 6 | 4 | 32 | 20 | 7160 |
| Beans | 1 kg | 588 | 213 | 12 | 2 | 218 | 0 | 3240 |
| Sugar | 1 kg | 1000 | 0 | 0 | 0 | 0 | 0 | 4000 |
| Corn Grits | 1 kg | 720 | 60 | 0 | 0 | 20 | 0 | 3200 |
| Fiber | 1 kg | 769 | 81 | 36 | 5 | 73 | 350 | 3620 |
| Pasta | 500 g | 376 | 63 | 6 | 0 | 0 | 0 | 1808 |
| Powder Milk | 400 g | 154 | 105 | 107 | 67 | 0 | 1484 | 1996 |
| Soybean oil | 900 mL | 0 | 0 | 831 | 138 | 0 | 0 | 7477 |
| Egg | 12 units | 9 | 70 | 48 | 14 | 0 | 907 | 772 |
| Total Kit | 9 Items | 5191 | 736 | 1046 | 230 | 343 | 2761 | 33,273 |
| Total/day | | 58 | 8 | 12 | 3 | 4 | 31 | 370 |

Source: Authors based on data from [34].

In total, 45,331 food kits were distributed at a unit cost of USD 6.90. The amount of food provided in each kit (33,273 kcal) means that, on average, the kit supplied more than 15% of the daily caloric needs of a child for 90 days, according to the objective of the program.

### 3.2. Purchase, Assemblage, and Distribution of Food Kits

To ensure urgent service to the program's beneficiaries, the Vitória Government established a deadline of 10 days for the first phase of the distribution of the food kits to all the schools. Between 21 July 2020 and 4 August 2020, the delivery of the kits to the beneficiaries was scheduled. SEME set a maximum deadline of three working days for the delivery of the kits.

The supply chain structure to distribute the food kits to the Vitória City Education Network worked according to the logistics process flow presented in Figure 4. One can observe the complexity of the logistics activity and verify that it covers the aspects of purchase, assembly, distribution, management, storage, and transportation.

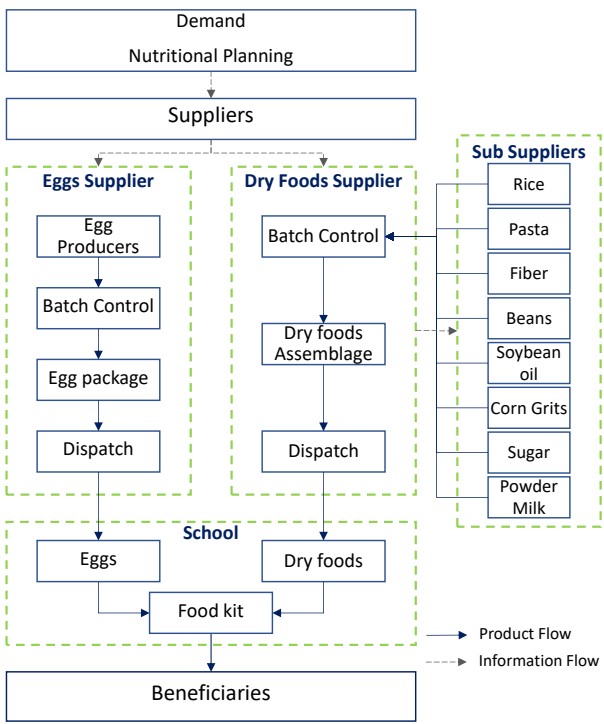

**Figure 4.** The structure of the supply chain of the food kit for Vitória City.

The demand was defined according to nutritional planning, and SEME, through the procurement team, carried out the bidding of the food kits.

To meet Brazilian regulations [12], one of the main challenges during the definition of the products that would compose the food kit distributed by SEME was the choice of food items from smallholder farmers.

Before the pandemic, the cooperative CAFSerrana, located 90 km from Vitória in the city of Santa Maria de Jetiba, Brazil's largest egg producer in 2019, already had a contract with the city of Vitória to supply fresh fruit, vegetables, and farm products from smallholder family farming. Through a survey conducted by SEME, during the selection of which items would come from smallholder farming, it was found that the existing contract between the municipality of Vitória and the cooperative could supply the number of eggs needed to meet the huge emergency demand. Thus, SEME chose this cooperative to supply the protein content and eggs as the protein choice for the food kits. According to this contract, the producers themselves delivered the products previously packed according to the needs of each school, aiming to facilitate the kit assemblage for food distribution.

Another factor that influenced the choice to include eggs in the kits was that, in the regions near the city of Vitória, there was large agricultural production of fresh products such as eggs and vegetables in smallholder family farming used to supply school meals in the pre-pandemic period. However, nonperishable products such as industrialized milk, rice, beans, flour, and oil are not in the small family farmer portfolio. Moreover, despite being an unprocessed product, eggs are a protein source and have a longer shelf life than vegetables, an essential factor due to the time required for the assemblage and distribution of food kits during the pandemic.

The selection of the dry food supplier, won by the company Vila Vitória, was done through a bidding process. This supplier had to execute all assembly and distribution of the nonperishable item kits to all schools. For the selection of the egg supplier, no bidding was necessary, as the current contract with CafSerrana Cooperative was used. CafSerrana collected eggs from local producers and performed batch quality verification, kit assembly, storage, and distribution to all schools.

Next, supplies from both suppliers were received and stored in the schools. In this process, the supplies went through visual screening. In the case of nonconformities, supplies were returned to the supplier. If approved, the supplies were delivered to beneficiaries (student families). The final kit was composed of the nonperishable products delivered from Vila Vitória plus the eggs supplied by CafSerrana.

*3.3. Survey Results*

Following the delivery of the food kits, a survey was conducted to understand whether the food kit delivered was able to fill the food necessities during the COVID-19 pandemic. The results provided a first indication of the cause-and-effect relationships in school food supply chains and the requirements for achieving program sustainability. The program was considered successful, as 96% of the beneficiaries received a kit, demonstrating that the program had an effective communication campaign, as 72% were informed through school communication channels and 19% through local media.

Highlighting the importance of the program to the students' families, approximately 87.5% of the beneficiaries reported that they suffered a financial impact on family income due to the lack of school meals during the pandemic.

One of the requirements for the delivery process was to avoid crowding in schools. It was observed that the delivery of the kits was relatively agile since 66.6% of beneficiaries were able to pick up their kit in under 10 min and only 26.5% had to stand in line.

For the quality of the products delivered, most beneficiaries were satisfied, since 97.1% of the beneficiaries received the kit in full and without any divergences or damage.

However, during the interview with SEME and the technical team, a problem with the visual aspect of the eggs delivered by the CafSerrana Cooperative was reported. The cooperative itself reported in its interview that 22% of the egg cardboard boxes had mold (nonconformity) and had to be returned, generating delays, rework, reverse logistics, and extra costs to the company. Since damaged products were replaced before distribution, the professionals did not report the problem significantly in the survey, and the beneficiaries did not even notice its occurrence.

Despite this success in supply chain management, 95.6% of the beneficiaries responded that the food in the kit was enough for less than one month (Figure 5). Most presumably, these families used the kit not as a food complement for their children but as the main source for the entire family.

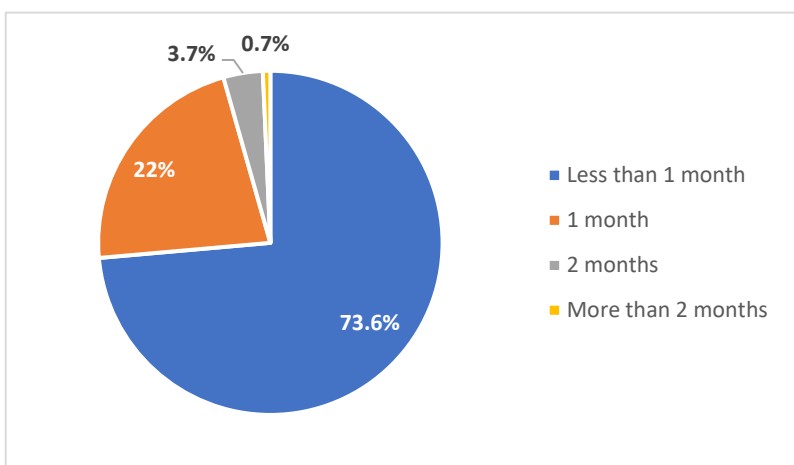

**Figure 5.** Food kit duration.

In the questionnaires for professionals (teachers and employees), there was a register for suggestions and improvements in the process. In this questionnaire, there were two categories of professionals, those who only worked in schools and those professionals with children enrolled as students in the schools. The suggestions are shown in Figure 6.

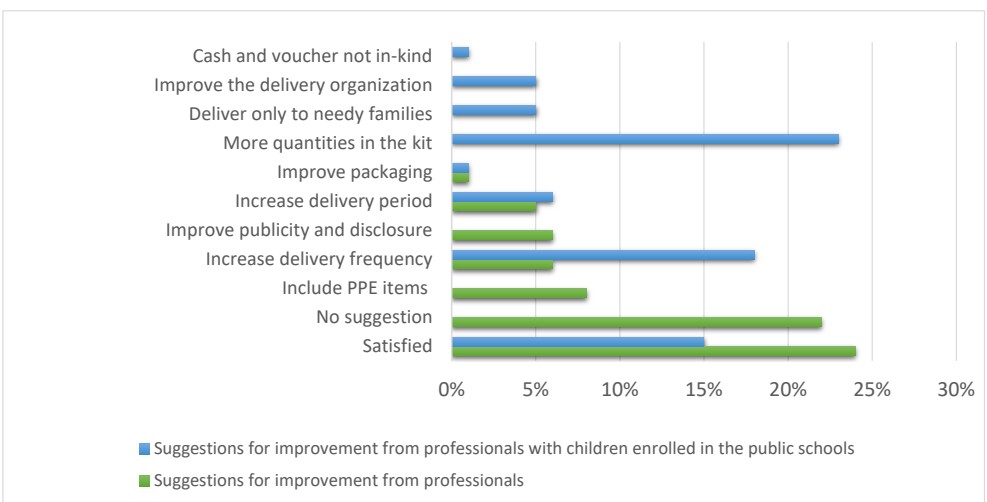

**Figure 6.** Professionals' suggestions for improvement.

Thus, from the supply chain mapping, the logistics flow presented satisfactory results among beneficiaries, professionals, and suppliers, and the suggestion for improvement with the highest percentage of answers was about increasing the kit size.

## 4. Discussion

There are many operational challenges related to the logistics of delivering products from suppliers to school units that directly interfere with product costs. One of the facilitating factors for the implementation of this project was the fact that the city government was part of the distribution, taking advantage of current logistics and adding fresh products delivered by local farmers.

In fact, the provision of school meals is one of the main public health and food and nutritional safety policies in Brazil, ensuring not only adequate and healthy food for children and youths but also contributing to the formation of good eating habits among students, which improves their performance and reduces school evasion.

In periods of normal school activities, the guidelines and norms established by the PNAE are fulfilled with few difficulties. A regular demand with weekly scheduled periodic deliveries of a variety of products from smallholder family farming, without a high variation in consumption, facilitates the logistics process.

However, during the period of social isolation due to the COVID-19 pandemic, the logistics process had to adapt in a short period of time to meet all PNAE execution legislation and ensure the continuity of the program.

Even with the publication of new regulations [7] that authorized the distribution of food kits purchased with PNAE funds, the greatest challenge faced in this period was to comply with the mandatory purchase of 30% of food from smallholder family farmers. The current legislation, designed to regulate the normal situation for the purchase of food from smallholder farms, created barriers in this process in emergencies, such as during the COVID-19 pandemic. The items selected to meet this requirement are usually fresh and highly perishable regional products, which are often incompatible with the time and process required to organize and distribute food kits.

Thus, the SEME technical team has struggled to define a smallholder family farming item to compose the kit. In the regions close to the city of Vitória, there is a large concentration of smallholder family farmers who cultivate fresh products and a reduced number who produce nonperishable products. This situation led to the choice of eggs, since, despite being an unprocessed product, this item has a longer shelf life than vegetables, which become unfit for consumption due to the longer time necessary for the assembly and distribution of food kits. The greatest challenge faced in the logistics process was

concerning the storage and perishability of eggs, in addition to handling issues due to their fragility.

During school closures, there was low demand for smallholder family farm items in fairs and street markets, which caused the eggs to be stored for a longer period. In addition, the city where the headquarters of the CafSerrana Cooperative is located is a mountainous region with a humid subtropical climate, wherein, in the period of intense cold, usually between May and August [35], producers suffer from the problem of moisture and mold storage, compromising the quality of eggs.

One alternative would be the purchase of less perishable products from smallholder farming, such as potatoes, beans, and rice; however, no local suppliers were found that could produce the necessary volume of the same product, a fact that would force the city to buy different items from different suppliers. However, it is recommended that the items should be the same for all beneficiaries, avoiding differences among food kits in the distribution. It is important to ensure that each beneficiary receives the product with the same description and packaging to prevent any kind of confusion or conflicts during distribution [36].

In the interviews carried out, it was observed that the difficulties faced by smallholder family farmers were as follows: (i) the region's producers were forced to keep an excessive amount of eggs stocked because of the reduced demand in street markets during the pandemic; (ii) the cooperative had to stock eggs until it reached the quantity needed to deliver to the school units, increasing again the storage time; (iii) the extension of the delivery date (anticipation of two weeks of school vacations) extended the storage time and aggravated the situation. It is believed that the sum of these storage times culminated in the nonconformity of approximately 22% of the product.

When questioned about the supply contracts of food kits, the president of the cooperative reported that the less perishable horticultural products, such as pumpkins, beets, carrots, chayotes, papayas, bananas, and onions, did not present deterioration or damage. However, he pointed out that it is necessary to organize and align the logistical structure to reduce the cycle of the logistics process and delivery of these products to less than two days.

It is of paramount importance to report that SEME immediately notified and returned the nonconforming eggs directly to the supplier; therefore, this problem was not reported in the beneficiaries' survey, which justifies the high satisfaction rate of the beneficiaries regarding the delivery of the kits.

Another positive point of this program, which should serve as a learning experience for future food distribution during crises, was the use of the existing logistics structure from end to end and the existence of suppliers who already knew all the regions, because the city of Vitória has a challenging topography, with hills, slopes, alleys, and stairways, that makes it difficult to access the schools. As reported by suppliers, the issue of knowing the obstacles of each region brought good results in the delivery of the kits.

It is also noteworthy that the suppliers of dry foods also had difficulties facing a scenario of a lack of food items available for purchase due to panic buying of these products at the beginning of the pandemic, which caused demand peaks for dry food items in supermarkets [37].

Although the main complaint by beneficiaries was linked to the food quantity and size of the kit, each food kit offered 370 kcal/day for 90 days per student. Considering that each student needs an average of 1885 kcal/day [31], in terms of energy value, the program met the goal of supplying a minimum of 15% of the nutritional needs of schoolchildren. However, it should be noted that the diversity and nutritional value of the products distributed, due to the new distribution model, is much lower than that offered daily at school.

The Brazilian regulations that determined the purchase of smallholder family farming products provided benefits in terms of nutrition and food quality, in addition to improving

the rural economy [8]; however, in times of crisis, it generated logistical difficulties, such as those faced by the program in the city of Vitória.

During the pandemic, the government was effective in changing the distribution of school meals for food kits purchased with PNAE funds. However, other aspects of the legislation, such as the mandatory purchase of local smallholder family farming products, should also be improved or made more flexible during periods of crisis to reduce costs and speed up the distribution of food to those who need it most.

## 5. Conclusions

The main objectives of this case study applied in the city of Vitória (Brazil) were to determine the difficulties in the acquisition, assembly, and distribution of food kits during the public schools' lockdown and the necessities for changing processes and legislation. The lessons learned allow for suggestions that would make this supply chain more resilient in future emergencies.

We mapped the process, conducted semi-structured interviews with professionals and food suppliers, and provided questionnaires to teachers, school workers, and parents. This research provided information about the logistics process, bottlenecks, failures, wastes, risks, and public regulations.

In the course of the study, the following difficulties were identified: (i) restrictions in the current legislation concerning school feeding; (ii) lockdown and social distancing; (iii) high quantity of products demanded in a short period; (iv) types of food produced by smallholder family agriculture in the region; (v) storage time needed due to the kit assemblage and distribution process; (vi) region's climate, which affects local production; and (vii) deficiency in infrastructure of the region's smallholder producers and cooperative.

It is noteworthy that most of the difficulties are related to the legislation in force, designed to regulate normal situations of school meal food acquisition, but which created obstacles in the supply chain of food kit distribution during the emergency closure of schools. Therefore, it is important to create coordinated actions, not only emergency actions but also others that also seek to mitigate the effects of crises, with medium- and long-term measures that can guarantee food security and the constitutional right to food.

As a scope for future research, we suggest the analysis of measures that encourage smallholder family farm producers to produce different foodstuffs, in addition to determining a logistics plan for emergencies. These measures may reduce the risk of food insecurity in children in a future emergency event.

As limitations of this case study, we note that (i) the case study was performed in the city of Vitória and can be only partially generalized to other cities in Brazil; (ii) we received questionnaire answers from all the schools in every region of Vitória; however, we do not have data about the number of answers of each school because SEME compiled these data; (iii) the quantities of calories needed for each student was determined as an average for all students; (iv) it was not our objective to discuss the legislation by itself.

**Supplementary Materials:** The following supporting information can be downloaded at: https://www.mdpi.com/article/10.3390/logistics6010020/s1.

**Author Contributions:** Conceptualization, T.M.C., J.Z.M. and I.d.B.J.; methodology, T.M.C., J.Z.M. and I.d.B.J.; data collection and processing, T.M.C. and J.Z.M.; formal analysis, T.M.C., J.Z.M., I.d.B.J. and A.A.M.; investigation, T.M.C., J.Z.M. and I.d.B.J.; writing—original draft preparation, T.M.C., J.Z.M., I.d.B.J., A.A.M. and H.T.Y.Y.; writing—review and editing, I.d.B.J., A.A.M. and H.T.Y.Y.; supervision, H.T.Y.Y.; project administration, I.d.B.J. All authors have read and agreed to the published version of the manuscript.

**Funding:** National Council for Scientific and Technological Research (CNPq), grant number 313687/2019-6.

**Institutional Review Board Statement:** Ethical review and approval were waived for this study due to anonymously answers of the questionnaires and that the first question was regarding acceptance of the informed consent form (ICF). The survey complied with all research-with-human rules from our institution.

**Informed Consent Statement:** Informed consent was obtained from all subjects involved in the study.

**Data Availability Statement:** Not applicable.

**Acknowledgments:** CNPq National Council for Scientific and Technological Research.

**Conflicts of Interest:** The authors declare no conflict of interest.

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
