# Peer review of "Public School Food Supply Chain during the COVID-19 Pandemic: A Case Study of the City of Vitória (Brazil)"

_logistics, 2021_

Round 1

Reviewer 1 Report

This is an exciting real-life problem, and there are many similar problems that we are not trying to solve through a systematic research method. Anyway, I see the absence of rigour in the methodology adopted.

This is a case study, whereas the abstract does not elaborate. Kindly rewrite the abstract with more focus on methodology and unique findings. Why should one read your paper?

The author should tell about the objectives of this research. In the last part of the paper should tell how much these objectives have been accomplished

Authors should provide the research method adopted in this paper; a small block diagram would be nice.

Need to provide limitations of this work before the last conclusion section

Scope for Future research and research implications are required as part of Conclusion subsection

Author Response

Response to Reviewer #1

This is an exciting real-life problem, and there are many similar problems that we are not trying to solve through a systematic research method. Anyway, I see the absence of rigour in the methodology adopted.

Authors: In Materials and Methods section, we reviewed and described the Case Study methodology based on the Eisenhardt (1989) seminal paper and Miguel (2007).

This is a case study, whereas the abstract does not elaborate.

Authors: We described in the Abstract that the paper is a case study.

Kindly rewrite the abstract with more focus on methodology and unique findings. Why should one read your paper?

Authors: We described in the Abstract that the paper is a case study and added the results of the case that allow to the suggestion of potential actions that would make this supply chain more resilient in future emergencies.

The author should tell about the objectives of this research. In the last part of the paper should tell how much these objectives have been accomplished.

Authors: Thank you very much for your comment! We reviewed Introduction and Conclusion according to the objectives.

Authors should provide the research method adopted in this paper; a small block diagram would be nice.

Authors: Thank you for the suggestion. In Material and Methods section, we inserted a flowchart illustrating the methodology. We also improved the description of the methodology.

Need to provide limitations of this work before the last conclusion section. Scope for Future research and research implications are required as part of Conclusion subsection

Authors: Thank you very much for your comment! We reviewed Conclusion and described the limitations and future research.

Reviewer 2 Report

The paper presents a very interesting case of food distribution due to COVID19 effect. It presents conclusions from of the questionnaires to users and experts, however it requieres a more detailed logistic and optimization analysis, in particular to identify models or techniques used or with high use potential.

Author Response

Response to Reviewer #2

The paper presents a very interesting case of food distribution due to COVID19 effect.

It presents conclusions from of the questionnaires to users and experts, however it requires a more detailed logistic and optimization analysis, in particular to identify models or techniques used or with high use potential.

Authors: Thank you very much for your comments! They helped us to greatly improve the paper. We appreciate all feedback and critics and tried our best to fulfill reviewers’ expectations. We improved the methodology defining the steps of the Case Study. The Reviewer will best identify that the case study does not aims any kind of optimization, but instead it aims to describe the lessons learned in the process of public-school food distribution system during the COVID-19 in the city of Vitoria (Brazil).

Reviewer 3 Report

In the abstract, please inform the purpose of the article, the research methodology, the main results, and the main implications of the results. In chapter 2 it is not necessary to use subsections. A single session is enough. Use bullets to identify steps in the methodology. After the bullets, in cursive text, describe the details of each step of the methodology. At the end of the chapter, inform what procedures were taken to ensure the validity and reliability of the findings. For this, consult the seminal article by Eisenhardt (1989) on how to conduct case studies. I don’t believe that food basket is a suitable term for what you’re intending to say. Please contact a native speaker for proofreading in the entire text. Please improve the quality of Figure 1. Currently, it is unreadable. There are no data on the sample of respondents of the survey. Please provide. The lack of a demographic analysis of the sample prevents the acceptance of the article. It lacks a more rigorous analysis of the results of the survey. Perhaps you should try regressions or probability distributions. It lacks a deeper analysis of the implications of the study. What is the main advantage entailed by the conclusions? Who wins something or benefits from the findings? Why a benefit exists? Please provide. You have a very serious problem with article references. References in Portuguese are not of interest to readers of an international newspaper. Replace all with English equivalents. Add more references, they are poor and few.

Author Response

Response to Reviewer #3

In the abstract, please inform the purpose of the article, the research methodology, the main results, and the main implications of the results.

Authors: We appreciate your comment! We described in the Abstract that the paper is a case study and added the results of the case that allow to the suggestion of potential actions that would make this supply chain more resilient in future emergencies.

In chapter 2 it is not necessary to use subsections. A single session is enough.

Authors: Thank you very much for your comment! We reviewed section 2 and it is now just a single section.

Use bullets to identify steps in the methodology. After the bullets, in cursive text, describe the details of each step of the methodology.

Authors: We rewrite the methodology according to the reviewer’s suggestion.

At the end of the chapter, inform what procedures were taken to ensure the validity and reliability of the findings. For this, consult the seminal article by Eisenhardt (1989) on how to conduct case studies.

Authors: Thank you very much for your comment and the suggested paper. It helped a lot. We rewrote the Materials and Methods section, following the seminal article by Eisenhardt (1989, according to the reviewer’s suggestion.

I don’t believe that food basket is a suitable term for what you’re intending to say.

Authors: Food basket is the term used in the Sphere Handbook that defines Standards in Humanitarian Response. We review the text and cite this document. Anyway, we changed the term food basket to food kit in order to be clearer.

Please contact a native speaker for proofreading in the entire text.

Authors: We reviewed the English but we didn´t use a proofreading service yet because the time to review was only 10 days. In case of acceptance, the paper will be reviewed by a proofreading service before final submission.

Please improve the quality of Figure 1. Currently, it is unreadable.

Authors: We reviewed and changed the layout of the figure.

There are no data on the sample of respondents of the survey. Please provide. The lack of a demographic analysis of the sample prevents the acceptance of the article. It lacks a more rigorous analysis of the results of the survey. Perhaps you should try regressions or probability distributions. It lacks a deeper analysis of the implications of the study.

Authors: In the topic Entering the field (Materials and Methods), we improved the description in order to clarify the sample characteristics. In Selecting the case (Materials and Methods), we also described some geographical and demographical aspects of Vitoria city.

What is the main advantage entailed by the conclusions? Who wins something or benefits from the findings? Why a benefit exists? Please provide.

Authors: We reviewed the conclusions in order to clarify the findings and the lessons learned.

You have a very serious problem with article references. References in Portuguese are not of interest to readers of an international newspaper. Replace all with English equivalents. Add more references, they are poor and few.

Authors: We reviewed the references and use the English version of the papers in Portuguese. We also added more references (from 21 to 36), mainly about the methodology and relevance of school feeding. The new references are identified in yellow.

Round 2

Reviewer 1 Report

I see authors have made improvements in the paper and the same can be accepted in the present form.

Author Response

Authors: Thank you very much for your opinion!

The changes are identified using Word Track Changes or highlighted in yellow.

Reviewer 2 Report

Suggestions and modifications solved adequately

Author Response

(The authors gave the same response as above.)

Reviewer 3 Report

authors have partially addressed some issues. A lack on the demography of two samples still remain, on the families that answer the questionnary and the professionals that answer questions. Another flaw is the lack of probability distributions of the answers of the surveys. Histgrams should help. Please provide. A proofreading is mandatory.

Author Response

Response to Reviewer #3

Reviewer comment: Authors have partially addressed some issues. A lack on the demography of two samples still remain, on the families that answer the questionnary and the professionals that answer questions.

Authors: We thank you for your recommendations. We have improved the part about the questionnaires samples (below) and added the questionnaires as Supplementary Material.

In the Brazilian society, usually high-income and mean-income students attend private schools, because it is known that private schools have a better education system. Public schools are usually attended by low-income students for who the school meals provided have a relevant importance in food security [25,26]. Figure 2 compares the monthly income of Vitoria city population and the public school’s student´s families according to Brazilian minimum wage (MW) (1 MW = 227.4 USD). It can be observed that more than 65% of the public-schools students’ families receive less than 2MW of monthly income. Also, mostly students of high-income families are not enrolled in public-schools.

Figure 2 Here

Figure 2. Comparison of family income of the public-school students’ families and the city population of Vitoria.

Source: [27]

According to IBGE (2019), 24.7% of Brazilian families are below the poverty line of 5.50 USD per day. This amount corresponds to less than 4/5 of Brazilian minimum wage per month (165 USD). Figure 2 shows that most of the students of these families are enrolled in public-schools.

Concerning the sample of the questionnaires sent to professionals, we received answers from professionals from all the school zones, as it can be seen in Figure 3. It is worth mentioning that different zones may have different quantities of public-schools and, consequently, different amounts of professionals.

Figure 3 Here

Figure 3. Amount of professional questionnaires answers from each school zone.

Source: Authors

                For the sample of the students, questionnaires did not collect information about school zones, because, according to the SEME managers, most of the parents or guardians could not be able to locate their school correctly. Also, it was not our objective to collected information about income, gender, or age since we know the income family profile of public-schools (Figure 2). Additionally, we were told by SEME that families of higher-income that have children enrolled in public-schools did not require the kit. Consequently, those questions were not an issue to our research.

Reviewer comment: Another flaw is the lack of probability distributions of the answers of the surveys. Histgrams should help. Please provide.

                Authors: We thanks the reviewer for this comment and think it would improve significantly our research. However, it is important to highlight that the answers for both the questionnaires collected attribute data to make the answers simple for people who may not have had the opportunity to get a high level of education. In this sense, data distribution probabilities are not possible to make. Please see both the questionnaires in Supplementary Material.

Reviewer comment: Please provide. A proofreading is mandatory.

Authors: The paper has been reviewed by a professional proofreading service. Attached certificate.

Round 3

Reviewer 3 Report

The authors addressed most issues. Histograms of the most important answers would heve helped to attract the interest of readers.